# Heart Team for Optimal Management of Patients with Severe Aortic Stenosis—Long-Term Outcomes and Quality of Life from Tertiary Cardiovascular Care Center

**DOI:** 10.3390/jcm10225408

**Published:** 2021-11-19

**Authors:** Szymon Jonik, Michał Marchel, Ewa Pędzich-Placha, Zenon Huczek, Janusz Kochman, Piotr Ścisło, Paweł Czub, Radosław Wilimski, Piotr Hendzel, Grzegorz Opolski, Marcin Grabowski, Tomasz Mazurek

**Affiliations:** 11st Department of Cardiology, Medical University of Warsaw, Banacha 1a Str., 01-267 Warsaw, Poland; szymonjonik.wum@gmail.com (S.J.); ewa.pedzich@wp.pl (E.P.-P.); zhuczek@wp.pl (Z.H.); j.kochman@tlen.pl (J.K.); piotr.scislo@wum.edu.pl (P.Ś.); grzegorz.opolski@wum.edu.pl (G.O.); marcin.grabowski@wum.edu.pl (M.G.); tomaszmazurek.wum@gmail.com (T.M.); 2Department of Cardiac Surgery, Medical University of Warsaw, Banacha 1a Str., 01-267 Warsaw, Poland; pczub@wum.edu.pl (P.C.); radoslaw.wilimski@wum.edu.pl (R.W.); phendzel@wum.edu.pl (P.H.)

**Keywords:** heart team, aortic stenosis, heart failure, transcatheter aortic valve replacement, surgical aortic valve replacement, optimal medical therapy

## Abstract

Background: This retrospective study was proposed to investigate outcomes of patients with severe aortic stenosis (AS) after implementation of various treatment strategies following dedicated Heart Team (HT) decisions. Methods: Primary and secondary endpoints and quality of life during a median follow-up of 866 days of patients with severe AS qualified after HT discussion to: optimal medical treatment (OMT) alone, OMT and transcather aortic valve replacement (TAVR) or OMT and surgical aortic valve replacement (SAVR) were evaluated. As the primary endpoint composite of all-cause mortality, non-fatal disabling strokes and non-fatal rehospitalizations for AS were considered, while other clinical outcomes were determined as secondary endpoints. Results: From 2016 to 2019, 176 HT meetings were held, and a total of 482 participants with severe AS and completely implemented HT decisions (OMT, TAVR and SAVR for 79, 318 and 85, respectively) were included in the final analysis. SAVR and TAVR were found to be superior to OMT for primary and all secondary endpoints (*p* < 0.05). Comparing interventional strategies only, TAVR was associated with reduced risk of acute kidney injury, new onset of atrial fibrillation and major bleeding, while the superiority of SAVR for major vascular complications and need for permanent pacemaker implantation was observed (*p* < 0.05). The quality of life assessed at the end of follow-up was significantly better for patients who underwent TAVR or SAVR than in OMT-group (*p* < 0.05). Conclusions: We demonstrated that after careful implementation of HT decisions interventional strategies compared to OMT only provide superior outcomes and quality of life for patients with AS.

## 1. Introduction

Aortic valve stenosis (AS) is the most widespread valvular heart disease (VHD) in the world and still remains the most common primary cause for valve surgery or catheter intervention in Europe and North America [1,2]. The increase in incidence of degenerative senile AS due to ageing of the population is expected. The prevalence of AS is relatively low among young and middle-aged (in the absence of a congenital abnormality), while its general presence is about 5% of the population at age of 65 years and 12.4% of elderly, affected 3.4% of severe AS in those aged 75 years and older, with this note, if patients with severe, symptomatic AS left untreated, have a limited life expectancy [3]. With a growing number of therapeutic options, an idea of multidisciplinary heart team (HT) for the management of individuals with complex diseases has been implemented and now plays a central concept in the care of patients with AS (class I recommendation) [1,2]. An approach of multidisciplinary experienced team, taking into account clinical, angiographic and echocardiographic data, risk stratification, long-term prognosis and patients preferences seems to be a rational tool, deciding on the best treatment method for these patients, burdened with many co-morbidities. The options for treating of severe symptomatic AS include surgical aortic valve replacement (SAVR)—the unique and undisputable reference procedure for many years, transcatheter aortic valve replacement (TAVR)—less invasive, but increasingly used for inoperable or prohibitive risk for surgery and optimal medical therapy (OMT) as a palliative care. As recommended by specialists in European and American guidelines for VHD, for all patients qualified for interventional treatment, the final decision between SAVR versus (vs.) TAVR should be considered by the HT after careful comprehensive evaluation of given individuals, favoring TAVR in elderly patients who are inoperable or at increased risk for surgery [1,2]. However, while the idea of HT is generally adopted in the medical society, no clear consensus on how HT should cooperate and what the desired goals are is established; most importantly, long-term results of HT decisions implementation and patients satisfaction are still poorly investigated. To our knowledge, some research papers regarding the influence of HT decisions on prognosis of AS-patients are available in the literature [4,5,6,7,8,9,10,11,12,13,14,15,16]; however, there are still few studies describing real-life HT cooperation, and more evidence investigating HT consistency and significance of decision making and performance on hard clinical endpoints are required. The purpose of this study is to evaluate AS-patients management, long-term outcomes and quality of life following HT decisions implementation in the daily clinical practice of a tertiary cardiovascular care center. We believe that the obtained results and conclusions we excogitated will be supportive for emphasizing the evidence-based role of HT in the decisions-making process for VHD-patients.

## 2. Methods and Study Design

This single-center cohort study was conducted in the 1st Department of Cardiology, Medical University of Warsaw, a large tertiary cardiovascular care center in Poland. A total number of 656 patients that consulted for severe symptomatic AS during 176 HT meetings in 2016–2019 were enrolled in the retrospective analysis. The inclusion criteria were: aged ≥18 years and complete clinical, echocardiographic and angiographic characteristics. The exclusion criteria included the following: pregnancy/lactation, disseminated neoplastic process and life expectancy <1 year. All of patients were evaluated in a weekly meeting by a HT composed of interventional cardiologists, cardiac surgeons, clinical cardiologists and non-invasive imaging specialists and qualified after HT discussion to one of three main strategies: OMT alone, OMT and TAVR or OMT and SAVR. Sequentially, 131 (20.0%) patients were excluded from further analysis due to: not meeting severe AS criteria, qualification for combined surgery—SAVR + mitral valve surgery or/and CABG (coronary artery bypass grafting), valve-in-valve intervention or active endocarditis—69, 43, 10, and 9 patients, respectively. The severe AS for all symptomatic patients in our study was defined as aortic valve area [AVA] < 1.0 cm^2^ (square centimeter) accompanying with aortic valve area indexed to BSA (body surface area) [AVA I] < 0.6 cm^2^/m^2^ (square centimeter per square meter), peak aortic-jet velocity (PAV) ≥ 4.0 m/s (meter per second) and mean aortic valve gradient (AVG) ≥ 40 mm Hg assessed by echocardiography (in accordance to ESC guidelines) [1]. Out of the 525 (80.0%) patients (54 (10.3%) cases re-discussed) that qualified for OMT, TAVR or SAVR, a total number of 43 (8.2%) participants were excluded due to: no consent with patient preference, loss of follow-up or death before implementation—21, 13, and 9 patients, respectively. Ultimately, in the final study, 482 (73.5%) patients with completely implemented HT decisions (OMT, TAVR, SAVR—79, 318, 85 patients, respectively) were included. OMT was defined as use of drugs as angiotensin-converting enzyme inhibitors (ACEI), angiotensin receptor blockers (ARB), angiotensin receptor-neprilysin inhibitors (ARNI), beta-blockers, digoxin, loop diuretics agents and aldosterone antagonists (MRA) in a manner that provides optimal reduction of the signs and symptoms of heart failure (HF) associated with aortic valve defect. The severity of HF symptoms was assessed using New York Heart Association (NYHA) classification, chronic kidney disease (CKD) defined as glomerular filtration rate (GFR) < 60 mL/min/1.73 m^2^ (millilitres per minute per 1.73 square meter), severe pulmonary arterial hypertension (PAH) as pulmonary artery systolic pressure (PASP) > 55 mmHg, anemia as hemoglobin level < 12 g/dL for women and <14 g/dL for men (g/dL—gram/decilitre), cancer—as active or up to 5 years back and smoking—as active or in the past. As the primary endpoint composite of all-cause mortality, non-fatal disabling strokes and non-fatal rehospitalizations due to AS at the end of follow-up (EOF) was considered, while assessed independently: the above three separately, cardiovascular (CV) death, non-fatal myocardial infarctions (MI), non-fatal strokes (disabling and non-disabling) for all strategies and new onset of atrial fibrillation (AF), acute kidney injury (AKI), infective valve endocarditis, need for permanent pacemaker implantation (PPI), major bleeding assessed by BARC (The Bleeding Academic Research Consortium) ≥ 3, major vascular complications and aortic valve re-interventions for TAVR and SAVR were only determined as secondary endpoints. All participants were observed for occurrence of endpoints with median follow-up of 866 (maximum—1824; minimum—365) days. The main outline of the study was presented in Figure 1. Additionally, general health status, using the short-form (SF)-36 questionnaire (totally and separately for physical component summary (PCS) and mental component summary (MCS)) before SAVR, TAVR and HT discussion (for patients qualified for OMT) and at the EOF for all alive participants (31 December 2020) was assessed. We have not yet obtained ethical/institutional review board (IRB) approval for our research, however, due to observational nature of the study, in accordance with applicable regulations, it is not required.

### Statistical Analysis

The PQStat software (version 1.6.6, PQStat, Poznań, Poland) was used for statistical analysis. The normality of distribution for continuous variables was confirmed with the Shapiro–Wilk test. Categorical data were expressed as counts and percentages, while continuous data were presented as mean ± SD. The comparison between groups of patients qualified for individual treatment strategies was performed using chi-square test and the statistical analysis was executed using 1-way analysis of variance (ANOVA). To compare the outcomes for all strategies with each other the hazard ratios (HRs) with 95% confidence intervals (95% CI) were calculated. To determine the independent predictors of outcomes in long term follow-up depending on the implemented HT-treatment strategy, multivariable, and multinominal logistic regression models were generated. Time to event analysis was performed using Kaplan–Meier curves. All *p* values (*p*) were given as least 2-sided, and *p* values lower than 0.05 were considered statistically significant.

## 3. Results

### 3.1. Study Population

From January 2016 to December 2019, 176 HT meetings were held, and a total of 482 patients with severe symptomatic AS meeting inclusion and exclusion criteria with completely implemented HT decisions (225 (46.7%) male, age (years, mean (SD)) = 78.1 (7.9), BMI [Body Mass Index] (kg/m^2^ [kilogram per square meter], mean (SD)) = 27.8 (4.9), 400 (83.0%) with symptoms of HF, NYHA (class, mean (SD)) = 2.35 (0.79), EuroSCORE II [European System for Cardiac Operative Risk Evaluation II] (%, mean (SD)) = 9.3 (9.7), STS score [Society of Thoracic Surgeons score] (%, mean (SD)) = 5.9 (2.0) and given co-morbidities) were followed up. The average delay from HT decision to implementation was (min–max): 39 (2–171) and 27 (2–58) days for TAVR and SAVR, respectively, *p* = 0.001. As regards statistically significant differences between TAVR, SAVR and OMT groups, patients qualified for OMT were older, more often frail, presented more often with HF, CAD, history of previous MI and percutaneous coronary intervention (PCI), stroke, peripheral artery disease (PAD), anemia, chronic pulmonary obstructive disease (COPD), cancer, more symptomatic (according to NYHA), with severe PH and more than moderate mitral regurgitation (MR) and tricuspid regurgitation (TR) assessed by echocardiography and with the highest risk of intervention assessed both by EuroSCORE II and STS score than those with implemented TAVR or SAVR, those qualified for TAVR had the highest BMI, were more often burdened with atrial fibrillation (AF), CAD and with history of pacemaker implantation (*p* < 0.05 for all). Baseline clinical characteristics (overall and by groups) in details was presented in Table 1.

### 3.2. Echocardiographic Parameters

All patients were assessed by echocardiography—from the OMT-group at the time of HT discussion and from the SAVR- and TAVR-groups before and after intervention (at the time of discharge from the hospital). Statistically significant differences in echocardiographic parameters before HT decisions implementation were observed in the following: ejection fraction of left ventricle (LVEF) and incidence of bicuspid valve with the highest in the SAVR-group and evaluation of severity of aortic stenosis assessed by AVA I with the lowest in the SAVR-group (*p* < 0.05 for all). The results of echocardiographic parameters of prosthetic valve assessed after SAVR or TAVR implementation differ between these two groups for LVEDD, Doppler velocity index (DVI) and peak AVG and were significantly better for SAVR-patients (*p* < 0.05 for all). The detailed echocardiographic results before and after implementing HT decisions were collected in Table 2.

### 3.3. Endpoints

The occurrence of the primary endpoint was statistically most frequent in OMT-group (75 patients (94.9%)), comparing to the SAVR and TAVR groups—28 (32.9%) and 110 (34.6%) patients, respectively (*p* < 0.05). Additionally, SAVR and TAVR were found to be significantly superior to OMT for all secondary endpoints (*p* < 0.05). Considering the endpoints for interventional strategies only—TAVR was associated with reduced risk of AKI (30 days), new onset AF (30 days and at the EOF) and major bleeding >3 according to BARC (30 days and at the EOF)—*p* < 0.05 for all. Conversely, the superiority of SAVR for major vascular complications (30 days) and need for PPI (30 days and at the EOF) was observed (*p* < 0.05 for all). However, no statistically significant differences between SAVR and TAVR for primary endpoint and other secondary endpoints were noticed. In-hospital mortality did not statistically differ between SAVR and TAVR strategy (6 (7.1%) vs. 20 (6.3%); *p* = 0.80), while length of stay (days (SD)) in the intensive care unit (ICU) was significantly longer for SAVR-patients (4.25 (3.7) vs. 1.83 (3.8); *p* < 0.05). The endpoints comparing TAVR, SAVR and OMT were detailed in Table 3. We also performed a subanalysis of patients with active or previous cancer qualified after HT evaluation to TAVR, SAVR or OMT. From the entire population of these individuals (*n* = 69; 14.3%), patients qualified for OMT had a significantly increased risk of all-cause mortality as compared with the TAVR and SAVR-groups (*p* < 0.01), while no differences in other outcomes were observed. Evaluating this subgroup for interventional strategies only, the higher incidence of major bleeding (EOF) in TAVR-patients vs. the SAVR-cohort was demonstrated (*p* < 0.01), while other endpoints did not differ significantly. The Kaplan–Meier curves comparing all strategies for primary and secondary endpoints were presented in Figure 2.

### 3.4. Quality of Life

Quality of life and general health status assessed before implementing HT decisions—PCS, MCS and total—did not statistically differ between treatment strategies (*p* > 0.01 for all). At the EOF the results of PCS, MCS and total for all alive participants were significantly the lowest for SAVR, then for TAVR and the highest for OMT-group (*p* < 0.05)—detailed in Table 4. Additionally, the subanalysis of patients with active or previous cancer demonstrated that for this subgroup patients qualified for OMT had noninferior initial, but significantly worse final (EOF) quality of life (total and components) as compared with interventional strategies (*p* < 0.01). No significant differences in quality of life before and after HT evaluation were observed between TAVR and SAVR. According to the Polish version of the questionnaire, with a maximum of 103 points for PCS and 68 points for MCS (171 points—total), the highest point value means the lowest quality of life assessment, while the lowest point value indicates the highest level of quality of life.

## 4. Discussion

Although risk assessment appears to be a crucial element in the appropriate preprocedural selection of the optimal management strategy for patients with AS, there are limitations to the scoring system used to estimate the risk of adverse outcomes and numerical other conditions should be evaluated to properly choose the best treatment option. Some studies suggest that the Heart Team approach may positively impact adherence to guideline-directed therapy, encourage the incorporation of patient preferences through the use of shared decision making, and improve overall outcomes [17,18]. There is recognition; however, for AS-patients, there are no randomized trials to support this approach; [19] rather, a single study describe outcomes of a multidisciplinary approach without a comparator [20].

In our opinion, only the cooperation of HT (where the risk assessment is only a component) provides a complex decision-making with appraisal of factors not routinely included in risk algorithms, which is the best to reflect the circumstances of real-world clinical practice. Although TAVR is an alternative to surgery in patients with severe symptomatic AS, there are still limited data on long-term clinical outcomes and bioprosthetic-valve function after TAVR as compared with SAVR [21,22,23]. Even more importantly, more clinical trials comparing treatment options for AS-patients mainly focus on interventional strategy and neglect the long-term outcomes and quality of life of patients enrolled to conservative management after HT evaluation. Only in few reports some data concerning this issue are available [8,9,14,15,16]. In our experience, such a cohort of patients consulting due to AS is quite large and the problem of their future care remains pressing. Hence, in our real-world clinical study, we decide to take into account all above lack in evidence regarding the importance of HT for AS-patients management.

Through our study, we would like to highlight the need for research to recognize the HT definition and range of functioning by which it can be assessed in order to advance our comprehension of the optimal care model for AS-patients. We emphasize that only HT with combination of many years’ clinical practice can lead to the long-term outcomes widely assigned to result from a HT approach. For now, the HT in the present sense is often cited as derived from two randomized trials comparing PCI and CABG in CAD—SYNTAX [24] and AS—PARTNER [25,26]. In these trials, a HT was used for selection of appropriate patients during eligibility screening. In the study of Leon MB et al. [25] for 358 patients with severe AS, who were no suitable candidates for surgery, TAVR was compared with standard medical therapy (including balloon aortic valvuloplasty) and at 1 year was demonstrated to significantly reduced the rates of overall mortality (30.7% vs. 50.7% with standard therapy; HR = 0.55, 95% CI 0.40–0.74, *p* < 0.001), the composite endpoint of overall mortality or rehospitalization (42.5% vs. 71.6% with standard therapy; HR = 0.46, 95% CI 0.35–0.59, *p* < 0.001), and HF symptoms—NYHA III or IV (25.2% vs. 58.0% with standard therapy, *p* < 0.001), despite the higher incidence of major strokes and major vascular events. Further [26], for high-risk patients at 1 year, TAVR was associated with similar rates of death from any cause (24.2% vs. 26.8%, *p* = 0.44), cardiac death (14.3% vs. 13.0%; *p* = 0.63), repeat hospitalization (18.2% vs. 15.5%, *p* = 0.38), major stroke (5.1% vs. 2.4%, *p* = 0.07) and MI (0.4% vs. 0.6%, *p* = 0.69) as compared to SAVR, while vascular complications were significantly higher in TAVI-group (18.0% vs. 4.8%, *p* < 0.001) and major bleeding in SAVR-group (25.7% vs. 14.7%, *p* < 0.001). At 1 year, no statistically significant differences in reducing of HF symptoms were observed between compared groups. However, the effectiveness of HT approach on clinical outcomes or life quality was not tested in these [25,26] trials. There are several methodological strengths in this study that reinforce the validity of the obtained results: all-comer nature, retrospective enrolment, systematic and meticulous patient assessment, complete median 2.5-years clinical follow-up, assessment of quality of life and the use of standardized definitions and endpoints for clinical outcomes. To highlighted benefits of our study: we assessed all treatment options for AS-management: TAVR, SAVR and OMT with the worst long-term outcomes in primary and secondary end-points for patients qualified for non-interventional treatment (OMT). Additionally, general health status, PCS and MCS were also the poorest for OMT-patients. These results emphasized the first and foremost implications of our study: for AS-patients the intervention (SAVR or TAVR) is without doubt worth considering and despite peri- and after-procedural complications could lengthen survival and improve quality of life. Moreover, quite a large group of patients (as regards the conditions of single-center study) and the median 2.5-years of follow up is sufficient to determine with high probability that decisions of our HT are adequate and consistent with clinical practice. Furthermore, properly selected endpoints, clearly reflecting the most common and serious complications of VHD-interventional treatment, prove the translatability of the obtained results on proper functioning of HT.

## 5. Conclusions

In this study, we raised the role of HT in decision-making process for patients with AS demonstrating that those qualified by our internal HT for interventional strategy achieved greater benefits in both endpoints and long-term quality of life as compared to pharmacological treatment only arm. These results require further confirmation in longer follow-up or multicenter studies and registers, but surely provide establishment of HT role both in clinical practice and guidelines for AS management.

## 6. Limitations

The main limitations of this study is its retrospective, non-randomized character and single-center design. Above that, the decision-making process must be assigned to our individual HT cooperation and cannot be considered as a general one. Additionally, proper and regular use of drugs by patients often remains a matter of trust, hence it is difficult to determine the credibility of the endpoints in the OMT-group. Moreover, patients with non-implemented decisions were not included into final analysis, so we do not have data of their follow-up.

## Figures and Tables

**Figure 1 jcm-10-05408-f001:**
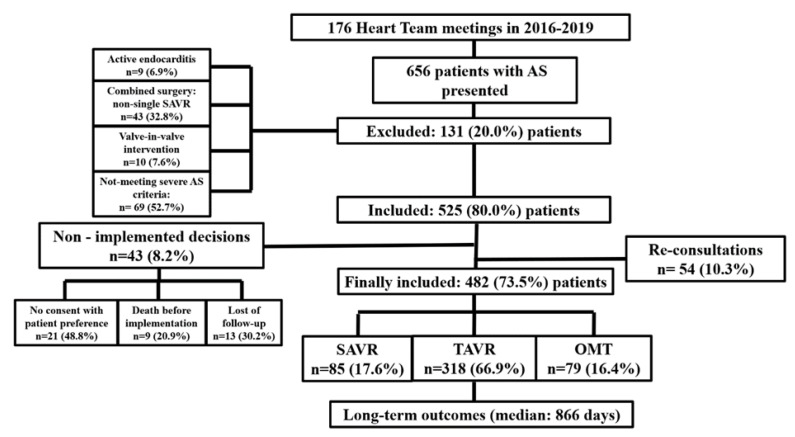
Study design. AS—aortic stenosis; TAVR—transcatheter aortic valve replacement; SAVR—surgical aortic valve replacement; OMT—optimal medical therapy.

**Figure 2 jcm-10-05408-f002:**
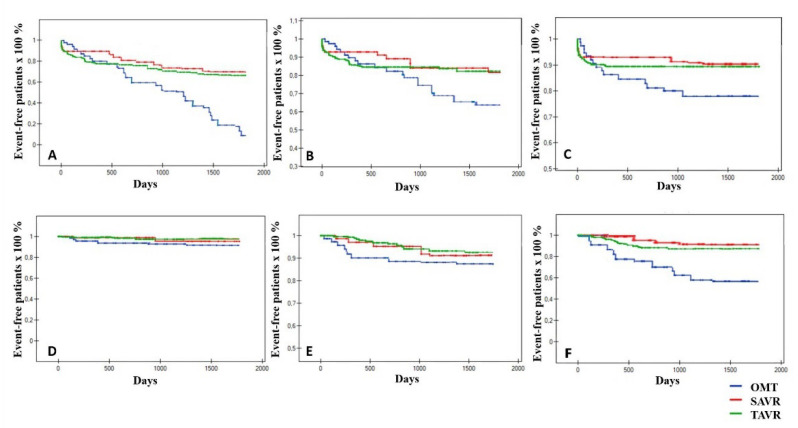
The Kaplan–Meier curves for endpoints. AS—aortic stenosis; TAVR—transcatheter aortic valve replacement; SAVR—surgical aortic valve replacement; OMT—optimal medical therapy; (**A**)—primary endpoint; (**B**)—all-cause mortality; (**C**)—CV (cardiovascular) death; (**D**)—non-fatal MI (myocardial infarction); (**E**)—non-fatal strokes; (**F**)—non-fatal rehospitalizations due to AS.

**Table 1 jcm-10-05408-t001:** Baseline clinical characteristics.

Baseline Characteristic	Overall (482)	TAVR (318)	SAVR (85)	OMT (79)	*p* Value
Age, years; mean (SD)	78.1 (7.9)	79.0 (7.2)	71.0 (6.2)	81.7 (8.0)	<0.01
Gender, male (%)	225 (46.7)	149 (46.9)	44 (51.8)	32 (40.5)	0.35
BMI, kg/m^2^; mean (SD)	27.8 (4.9)	28.2 (4.8)	27.9 (3.9)	26.4 (5.9)	0.01
Heart Failure, *n* (%)	400 (83.0)	279 (87.7)	50 (58.8)	71 (89.9)	<0.01
NYHA, mean (SD)	2.35 (0.79)	2.28 (0.71)	2.17 (0.77)	2.84 (0.93)	<0.01
CAD, *n* (%)	284 (58.9)	188 (51.9)	39 (45.9)	57 (72.2)	<0.01
Diabetes, *n* (%)	176 (36.5)	120 (37.7)	23 (27.1)	33 (41.8)	0.11
Hypertension, *n* (%)	417 (86.5)	276 (86.8)	72 (84.7)	69 (87.3)	0.86
Previous stroke/TIA, *n* (%)	47 (9.8)	30 (9.4)	4 (4.7)	13 (16.5)	0.04
Atrial fibrillation, *n* (%)	171 (35.5)	122 (38.4)	19 (22.4)	30 (38.0)	0.02
Previous MI, *n* (%)	99 (20.5)	49 (15.4)	13 (15.3)	37 (46.8)	<0.01
Previous PCI, *n* (%)	246 (51.0)	167 (52.5)	30 (35.3)	49 (62.0)	<0.01
Previous CABG, *n* (%)	53 (11.0)	29 (9.1)	10 (11.8)	14 (17.7)	0.09
Previous non-aortic VS, *n* (%)	25 (5.2)	14 (4.4)	3 (3.5)	8 (10.1)	0.09
History of pacemaker, *n* (%)	71 (14.7)	57 (17.9)	1 (1.2)	13 (16.5)	<0.01
≥Moderate MR, *n* (%)	26 (5.4)	11 (3.5)	3 (3.5)	12 (15.2)	<0.01
≥Moderate TR, *n* (%)	26 (5.4)	14 (4.4)	1 (1.2)	11 (13.9)	<0.01
PAD, *n* (%)	86 (17.8)	48 (15.1)	9 (10.6)	29 (36.7)	<0.01
CKD, *n* (%)	303 (62.9)	223 (70.1)	31 (36.5)	49 (62.0)	<0.01
Anemia, *n* (%)	289 (60.0)	206 (64.8)	25 (29.4)	58 (73.4)	<0.01
Dyslipidemia, *n* (%)	371 (77.0)	248 (78.0)	64 (75.2)	59 (74.7)	0.76
COPD, *n* (%)	72 (14.9)	42 (13.2)	6 (7.1)	24 (30.4)	<0.01
Severe PH, *n* (%)	46 (9.5)	21 (6.6)	3 (3.5)	22 (27.8)	<0.01
Cancer, *n* (%)	69 (14.3)	39 (12.3)	9 (10.6)	21 (26.6)	<0.01
Smoking, *n* (%)	323 (67.0)	216 (67.9)	53 (62.4)	54 (68.4)	0.6
Frailty, *n* (%)	221 (45.9)	151 (47.5)	11 (12.9)	59 (74.7)	<0.01
EuroSCORE II, %; mean (SD)	9.3 (9.7)	9.7 (11.0)	5.7 (4.0)	11.4 (7.3)	<0.01
STS score, %; mean (SD)	5.9 (2.0)	6.0 (2.1)	3.6 (1.7)	7.1 (2.6)	<0.01

Abbreviations. TAVR—transcatheter aortic valve replacement; SAVR—surgical aortic valve replacement; OMT—optimal medical therapy; BMI—body mass index, NYHA—New York Heart Association; CAD—coronary artery disease; TIA—transient ischemic attack, MI—myocardial infarction; PCI—percutaneous coronary intervention; CABG—coronary artery bypass grafting; VS—valvular surgery; MR—mitral regurgitation; TR—tricuspid regurgitation; PAD—peripheral artery disease, CKD—chronic kidney disease; COPD—chronic obstructive pulmonary disease; PH—pulmonary hypertension; EuroSCORE II—European System for Cardiac Operative Risk Evaluation II; STS score—Society of Thoracic Surgeons score.

**Table 2 jcm-10-05408-t002:** Echocardiographic parameters before and after Heart Team (HT) decisions implantation.

	Baseline					At Discharge				
Echocardiographic Parameter	Overall (483)	TAVR (318)	SAVR (85)	OMT	*p* Value	Overall (377)	TAVR (298)	SAVR (79)	OMT (79)	*p* Value
LVEF, %; mean (SD)	52.5 (13.4)	54.2 (12.5)	56.5 (12.1)	41.6 (12.5)	<0.01	55.9 (10.6)	55.4 (10.9)	57.3 (9.9)	-	0.2
LVEDD, cm; mean (SD)	5.3 (0.78)	5.2 (0.76)	5.3 (0.82)	5.4 (0.79)	0.16	5.2 (0.80)	5.2 (0.78)	5.0 (0.85)	-	0.03
LVESD, cm; mean (SD)	3.2 (0.79)	3.2 (0.77)	3.2 (0.81)	3.3 (0.82)	0.63	3.2 (0.76)	3.2 (0.76)	3.2 (0.78)	-	0.76
IVSD, cm; mean (SD)	1.36 (0.18)	1.35 (0.18)	1.39 (0.16)	1.33 (0.19)	0.11	1.33 (0.17)	1.33 (0.18)	1.34 (0.16)	-	0.76
AVA, cm^2^; mean (SD)	0.78 (0.15)	0.79 (0.16)	0.76 (0.14)	0.77 (0.13)	0.25	1.78 (0.40)	1.77 (0.41)	1.82 (0.37)	-	0.44
AVA I, cm^2^/m^2^; mean (SD)	0.47 (0.18)	0.45 (0.16)	0.49 (0.18)	0.53 (0.20)	<0.01	0.97 (0.24)	0.96 (0.25)	0.99 (0.18)	-	0.51
PAV, m/s; mean (SD)	4.38 (0.94)	4.34 (0.94)	4.59 (0.97)	4.32 (0.91)	0.07	2.23 (0.79)	2.27 (0.74)	2.16 (0.89)	-	0.19
DVI; mean (SD)	0.20 (0.05)	0.20 (0.05)	0.21 (0.04)	0.20 (0.06)	0.39	0.41 (0.09)	0.40 (0.09)	0.45 (0.08)	-	<0.01
Peak AVG, mmHg; mean (SD)	68.6 (24.5)	69.2 (26.4)	70.3 (20.7)	64.2 (19.6)	0.21	20.3 (7.6)	21.2 (7.4)	17.3 (7.5)	-	<0.01
Mean AVG, mmHg; mean (SD)	42.6 (16.3)	42.3 (16.4)	45.4 (12.6)	41.0 (19.2)	0.18	12.4 (6.0)	12.7 (6.0)	11.3 (5.7)	-	0.07
Bicuspid valve, *n* (%)	40 (8.3)	20 (6.3)	13 (15.3)	7 (8.9)	0.03	-	-	-	-	-
≥Moderate PVAR, *n*/all (%)	-	-	-	-	-	5 (1.3)	5 (1.7)	0 (0.0)	-	0.25
≥Moderate total AR, *n*/all (%)	-	-	-	-	-	9 (2.4)	8 (2.7)	1 (1.3)	-	0.46

Abbreviations. TAVR—transcatheter aortic valve replacement; SAVR—surgical aortic valve replacement; OMT—optimal medical therapy; LVEF—left ventricular ejection fraction; LVEDD—left ventricular end-diastolic diameter; LVESD—left ventricular end-systolic diameter; IVSD—intraventricular septum diameter; AVA—aortic valve area; AVA I—indexed aortic valve area; PAV—peak aortic velocity; DVI—doppler velocity index; AVG—aortic valve gradient; PVAR—paravalvular aortic regurgitation; AR—aortic regurgitation.

**Table 3 jcm-10-05408-t003:** Primary and secondary endpoints.

Endpoints	TAVR (318)	SAVR (85)	OMT (79)	*p* Value Overall	TAVR vs. SAVR HR [95% CI]; *p*	SAVR vs. OMT HR [95% CI]; *p*	TAVR vs. OMT HR [95% CI]; *p*
Primary Endpoint, *n* (%)	110 (34.6)	28 (32.9)	75 (94.9)	<0.01	1.05 [0.73–1.51]; 0.78	0.35 [0.22–0.55]; < 0.01	0.36 [0.25–0.53]; <0.01
Secondary Endpoints, *n* (%)							
All-cause mortality	50 (15.7)	14 (16.5)	29 (36.7)	<0.01	0.95 [0.55–1.65]; 0.88	0.45 [0.22–0.90]; <0.01	0.43 [0.24–0.75]; <0.01
Non-fatal strokes	18 (5.7)	7 (8.2)	11 (13.9)	0.04	0.69 [0.29–1.65]; 0.38	0.59 [0.19–1.81]; 0.25	0.41 [0.17–1.00]; 0.01
Non-fatal disabling strokes	9 (2.8)	4 (4.7)	8 (10.1)	0.02	0.60 [0.19–1.89]; 0.39	0.46 [0.11–2.02]; 0.19	0.28 [0.09–0.91]; <0.01
CV death	36 (11.3)	9 (10.6)	19 (24.1)	<0.01	1.07 [0.55–2.06]; 0.85	0.44 [0.19–1.02]; 0.02	0.47 [0.24–0.93]; <0.01
Non-fatal MI	9 (2.8)	3 (3.5)	8 (10.1)	0.01	0.80 [0.25–2.60]; 0.74	0.35 [0.08–1.57]; 0.09	0.28 [0.08–0.94]; <0.01
AKI (30 days)	9 (2.8)	12 (14.1)	-	-	0.20 [0.07–0.57]; <0.01	-	-
New onset AF (30 days)	24 (7.5)	13 (15.3)	2 (2.5)	<0.01	0.49 [0.21–1.14]; 0.03	6.04 [2.06–17.73]; <0.01	2.98 [1.25–7.09]; 0.11
New onset AF (EOF)	38 (11.9)	19 (22.4)	10 (12.7)	0.05	0.53 [0.28–1.02]; 0.01	1.77 [0.78–4.02]; 0.11	0.94 [0.49–1.83]; 0.86
Major bleeding (30 days)	19 (6.0)	11 (12.9)	-	-	0.46 [0.19–1.11]; 0.02	-	-
Major bleeding (EOF)	36 (11.3)	17 (20.0)	1 (1.3)	<0.01	0.57 [0.28–1.16]; 0.04	15.80 [6.33–39.45]; <0.01	8.94 [4.28–18.67]; <0.01
Major vascular complications (30 days)	32 (10.06)	2 (2.35)	-	-	4.28 [2.21–8.29]; 0.02	-	-
Infective valve endocarditis (30 days)	5 (1.6)	1 (1.2)	-	-	1.33 [0.19–9.50]; 0.79	-	-
Infective valve endocarditis (EOF)	8 (2.5)	3 (3.5)	3 (3.8)	0.78	0.71 [0.18–2.90]; 0.61	0.93 [0.15–5.61]; 0.93	0.66 [0.16–2.81]; 0.54
Permanent pacemaker implantation (30 days)	53 (16.7)	6 (7.1)	-	-	2.36 [1.26–4.41]; 0.03	-	-
Permanent pacemaker implantation (EOF)	72 (22.6)	9 (10.6)	6 (7.6)	0.01	2.14 [1.22–3.76]; 0.01	1.39 [0.68–2.87]; 0.51	2.98 [1.67–5.32]; <0.01
Aortic valve re-interventions (EOF)	6 (1.9)	2 (2.4)	-	-	0.80 [0.15–4.38]; 0.78	-	-
Non-fatal rehospitalizations for AS (EOF)	51 (16.0)	10 (11.8)	38 (48.1)	<0.01	1.36 [0.80–2.31]; 0.33	0.24 [0.12–0.48]; <0.01	0.33 [0.19–0.57]; <0.01
In-hospital mortality	20 (6.3)	6 (7.1)	-	-	0.89 [0.35–2.29]; 0.8	-	-
ICU stay, days (SD)	1.8 (3.8)	4.2 (3.7)	-	-	<0.01	-	-

Abbreviations. AS—aortic stenosis; TAVR—transcatheter aortic valve replacement; SAVR—surgical aortic valve replacement; OMT—optimal medical therapy; CV—cardiovascular; MI—myocardial infarction; AKI—acute kidney injury; AF—atrial fibrillation; ICU—intensive care unit; EOF—end of follow-up.

**Table 4 jcm-10-05408-t004:** The quality of life before and after Heart Team (HT) decisions implementation.

	TAVR (318/268)	SAVR (85/71)	OMT (79/50)	*p*-Value
Physical Component Summary (PCS)				
Before SAVR, TAVR, HT disscusion; mean (SD)	81.5 (14.4)	79.2 (16.5)	83.1 (13.0)	0.22
After SAVR, TAVR, HT disscusion—at the end of follow up; mean (SD)	69.5 (13.9)	65.7 (16.1)	84.1 (12.5)	<0.01
Mental Component Summary (MCS)				
Before SAVR, TAVR, HT disscusion; mean (SD)	52.9 (8.6)	52.1 (9.8)	53.8 (7.8)	0.46
After SAVR, TAVR, HT disscusion—at the end of follow up; mean (SD)	41.4 (8.3)	39.1 (9.2)	56.5 (8.0)	<0.01
Total				
Before SAVR, TAVR, HT disscusion; mean (SD)	134.5 (17.5)	131.3 (20.2)	136.9 (13.8)	0.12
After SAVR, TAVR, HT disscusion—at the end of follow up; mean (SD)	111.0 (16.9)	104.8 (21.0)	140.6 (13.5)	<0.01

Abbreviations. TAVR—transcatheter aortic valve replacement; SAVR—surgical aortic valve replacement; OMT—optimal medical therapy; HT—Heart Team.

## Data Availability

The data presented in this study are available on request from the corresponding author. The data are not publicly available due to any accessible repository.

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
