# Peer review of "Heart Team for Optimal Management of Patients with Severe Aortic Stenosis—Long-Term Outcomes and Quality of Life from Tertiary Cardiovascular Care Center"

_jcm, 2021, doi:10.3390/jcm10225408_

Round 1

Reviewer 1 Report

The present study sought to identify the importance of having a multidisciplinary heart team to help make decisions regarding the treatment of severe AS – OMT, TAVR, or SAVR. The authors included clinical endpoints and complications, but they particularly looked at quality of life as well.

Well designed study, well written. I applaud the authors for including active and history of cancer in their patient population as well, I'm curious whether there were any differences in quality of life or other endpoints in that population group as well. 

I did appreciate the comparison between the quality of life between the treatment groups, which is interesting and novel. I would suggest using abbreviations consistently (MHT vs HT) and maybe shortening some sentences where able (to make the text more easily readable). 

Author Response

Dear

We greatly appreciate you having reviewed our manuscript.

Below we presented our responses to your comments.

1. About what you asked: any differences in quality of life or other endpoints in population with active or history of cancer, we added in the text:

1) Section 3.3. Endpoints. Page 7, line 201-208:

" We also performed a subanalysis of patients with active or previous cancer qualified after HT evaluation to TAVR, SAVR or OMT. From the entire population of these individuals (n=69; 14.3 %), patients qualified for OMT had significantly increased risk of all-cause mortality as compared with TAVR and SAVR-group (P<0.01), while no differences in other outcomes were observed. Evaluating this subgroup for interventional strategies only, the higher incidence of major bleeding (EOF) in TAVR-patients vs SAVR-cohort was demonstrated (P<0.01), while other endpoints did not differ significantly. "

2) Section 3.3. Quality of life. Page 9, line 225-229:

" Additionally, the subanalysis of patients with active or previous cancer demonstrated that for this subgroup patients qualified for OMT had noninferior initial but significantly worse final (EOF) quality of life (total and components) as compared with interventional strategies (P<0.01). No significant differences in quality of life before and after HT evaluation were observed between TAVR and SAVR. "

2. We standarized the nomenclature - all 'MHT' abbreviations we changed into 'HT' abbreviations to make the text more understandable.

Thank you for taking the time for review this article.

With best regards

Tomasz Mazurek, MD, PhD

1st Department of Cardiology

Medical University of Warsaw,

Banacha 1a Street

01-267 Warsaw, Poland

e-mail: tommazurek.wum@gmail.com

tel.: + 48 22 599 19 58

fax.: + 48 22 599 19 57

Reviewer 2 Report

Dear Authors,

Thank you for allowing me to revise the fascinating manuscript entitled: Heart Team for optimal management of patients with severe aortic stenosis - long-term outcomes and quality of life from tertiary cardiovascular care center.

The manuscript deals with a crucial topic in the decision-making for patients affected by severe aortic stenosis. The evolution toward a holistic evaluation of the patients undergoing assessment for the best aortic stenosis treatment represents a mainstream route toward personalized medicine. The weight of frailty in decision-making is one of the most intriguing data, although the definition and measurement of frailty represent a sort of holy grail.

The manuscript fairly treats the topic and reports exciting data regarding the impact of the decisions of a multidisciplinary heart team on the outcome and quality of life of the patients discussed.

The data are of significant interest; however, the manuscript doesn't permit the reader to understand the reasons for a decision over another. 

I suggest the authors also use the STS score to predict morbidity and prolonged ICU stay since I expect these parameters to play a significant role in the decision-making of those patients. Please provide these data as they could possibly guide the clinician in the difficult choice of the best procedure on the basis of preoperative features.

Furthermore, I suggest adopting (if the data are available) also the Days Alive Out of Hospital as a simple and effective parameter to measure the impact of the selected approach for treating aortic stenosis.

My best regards

Author Response

Dear

We greatly appreciate you having reviewed our manuscript.

Below we presented our responses to your comments.

  1. About what you asked: STS score, we added in the text:

1) Section 3.1. Study population. Page 4, line 146-147:

" STS score [Society of Thoracic Surgeons score] (%, mean (SD)) = 5.9 (2.0) "

2) Section 3.1. Study population. Page 4, line 156:

" and STS score "

3) Table 1. Baseline clinical characteristics. Page 5, the last line: "

STS score, %; mean (SD)

5.9 (2.0)

6.0 (2.1)

3.6 (1.7)

7.1 (2.6)

<0.01

"

4) Table 1. Abbreviations. Page 5, line 167:

" STS score – Society of Thoracic Surgeons score. "

  1. We regret to say we have no data about “Days Alive Out of Hospital”. We agree this one will be a simple and effective parameter to reflect the impact of the HT approach for aortic stenosis outcomes, but we have not been able to collect this data. We are committed to providing such data for future articles evaluating HT management.

Thank you for taking the time for review this article.

With best regards

Tomasz Mazurek, MD, PhD

1st Department of Cardiology

Medical University of Warsaw,

Banacha 1a Street
01-267 Warsaw, Poland
e-mail: tommazurek.wum@gmail.com
tel.: + 48 22 599 19 58
fax.: + 48 22 599 19 57
